# Plant Recovery after Metal Stress—A Review

**DOI:** 10.3390/plants10030450

**Published:** 2021-02-27

**Authors:** Jagna Chmielowska-Bąk, Joanna Deckert

**Affiliations:** Department of Plant Ecophysiology, Institute of Experimental Biology, Faculty of Biology, Adam Mickiewicz University, Poznań, ul. Uniwersytetu Poznańskiego 6, 61-614 Poznań, Poland

**Keywords:** cadmium, lead, copper, zinc, toxicity, genotoxicity, recovery

## Abstract

Contamination of the environment with metals, their adverse impact on plant performance and transmission to the human food chain through crops and vegetables are important concerns worldwide. Although the literature on metal contamination, toxicity and plant response to this stress factor is quite abundant, there are very limited reports on the phenomenon of plant recovery after metal stress. The present article reviews available literature on the recovery process examined in various plant species, in response to several metals (Al, Cd, Cu, Ni, Pb, Zn), applied at different concentrations and treatment duration. The reviewed studies have been carried out in laboratory conditions. However, it should be highlighted that although metal stress is not as transient as most of other stress factors (e.g., drought, heat, chilling), metal concentration in the soil may still decrease due to, e.g., leaching to lower soil layers or uptake by organisms. Thus, in natural conditions, plants may be subjected to post-metal-stress conditions. The review also discusses the mechanism behind efficient recovery and the impact of post metal stress on future plant performance—possible acquisition of stress memory, adaptation to unfavorable conditions and cross-tolerance towards other stress factors.

## 1. Introduction

Contamination of the environment with metals/metalloids is one of the important concerns worldwide. Elevated levels of these elements in the soil have been detected in various regions of the world. The land use and cover area frame statistical topsoil survey (LUCAS) revealed high concentrations of Cd, Pb, Hg and As in industrious areas in Europe including Southern Saxony and the Ruhr region in Germany, Southern Poland, Central Rumania, Northern Spain and Lyon and Nimes regions in France [1]. Contamination of water and soil has also become increasing problem on other continents. In Africa, the most widespread contaminants include Cd and Pb, although other metals/metalloids such as Hg and As are also detected in specific locations. The pollution of the environment resulted in metal accumulation in crops, fish, cattle and eventually humans [2]. A comprehensive review of data on the contamination of urban soils in China showed that the level of all eight analyzed metals/metalloids (As, Cd, Cr, Hg, Pb, Cu, Zn and Ni) exceeded the background levels, with Cd and Hg being the most significant pollutants [3]. Metal pollution is associated with a decrease in soil fertility and biodiversity resulting in plant nutrient deficiencies and decreased crop yield [4]. In addition, metals/metalloids taken up by plants negatively affect their growth and fitness through disturbances in, among others, ultrastructure, mineral homeostasis, cell division and photosynthesis (described with references in Section 2). Several reports emphasize that metals/metalloids accumulated in crops pose a threat for human health. Elevated levels of metals, including As, Cd, Cr, and Pb, were detected in staple crops (e.g., rice and maize) and vegetables. In some of the studies, the level of metals exceeded permissible norms by 10 to 15 times [5,6]. The consumption of contaminated edible plants may lead to severe consequences as metals/metalloids exhibit neurotoxic, nephrotoxic, hematotoxic, hepatotoxic, genotoxic and carcinogenic effects in humans [5,6,7].

Due to all of the above-mentioned facts, the topic of metal stress is constantly in the center of attention of plant biologists, which is reflected in nearly 1000 articles published within the past five years containing terms “metal stress” and “plant*” in the title, abstract or as keywords (search in Scopus database, date of access: 14 February 2021). Most of the articles focus on metal accumulation, toxicity and plants’ defense response. In contrast, there are very limited studies on the process of post-stress recovery. Metal stress is less transient than other stresses such as high and low temperature, drought or flooding. However, metal concentrations in the soil might be decreased with time due to various processes including uptake by plants and other organisms, soil erosion by wind and water or leaching to lower soil layer. Therefore, plants can enter a post-metal-stress stage also in natural conditions [7]. Examination of the events following stress attenuation is important for the proper evaluation of stress tolerance. Tolerance is usually estimated on the basis of changes in growth parameters measured straight after exposure to unfavorable conditions. However, some plant species have adapted a quiescence strategy; they hamper growth during stress to save energy (carbohydrates reserve) for fast growth restoration after return to optimal conditions. In contrast, other species continue to grow at a relatively normal pace. These species are more likely to deplete their energy and nutrient supplies and perform less efficiently after stress termination. Therefore, the commonly applied methods for the estimation of tolerance directly after exposure to unfavorable conditions might provide an inexact view. It is suggested to include the recovery phase in the tolerance tests. Otherwise, the results might present a misleading view on plants’ real tolerance and regeneration capabilities after stress conditions [8].

The aim of the present article is to review the information on plant recovery after metal stress. The first section briefly describes mechanisms of metal toxicity. The second part of the article is an overview of studies on plant recovery from metal stress and possible mechanisms, which stand behind efficient regeneration after metal treatment. The impact of past stress on future plant performance is discussed. It is highlighted that past exposure to metals might leave relatively stable imprints in plants affecting their future performance and stress response.

## 2. Metal Toxicity in Plants

The adverse impact of metals/metalloids on plants is well-documented and comprehensively reviewed in several publications [9,10,11]. Thus, this section will only briefly point out the most important molecular and physiological mechanisms of metal toxicity and its effect on plants. Metals/metaloids can bind directly with protein functional groups (e.g., thiol, imidazole, carboxyl groups) leading to changes in their conformation and functioning. These elements can also affect protein folding and refolding processes, which results in the formation of misfolded, non-functional proteins or protein aggregates [12]. Metals might displace other elements in biomolecules. For instance, replacement of Ca by Cd in radish calmodulin resulted in its reduced activity [13]. Several metals were also shown to substitute Mg in chlorophyll, negatively affecting the process of photosynthesis [14]. The indirect mechanisms of metal/metalloid toxicity includes the overaccumulation of reactive oxygen species (ROS) leading to oxidative stress. High levels of ROS cause oxidation of biomolecules including membrane lipids, proteins, carbohydrates and nucleic acids. This, in turn, results in membrane damage, electrolyte leakage, increased mutation rate and reduced efficiency of various metabolic processes [15].

Metal stress is frequently accompanied by water deficit [16]. The mechanisms of metal-dependent dehydration were reviewed in detail by Rucińska-Sobkowiak [16]. In short, water uptake from the soil might be hampered due to reduced root growth and decreased root hair density [16] and referenced therein. Short-distance water transport is most likely inhibited by changes in the expression of aquaporins and an increase in cell wall thickness and deposition of callose [17,18,19,20]. Alterations in long distance transport are associated with a reduction of the size of vessels and increased deposition of debris and gums [21,22,23]. Metals can interact with the uptake of other elements from the soil. These interactions can be of a synergistic or antagonistic nature [24,25,26,27,28,29,30]. Antagonistic interactions lead to decreased uptake of certain minerals and nutrient deficiencies. For instance, *Aeluropus littoralis* plants exposed to Cd exhibited a hampered accumulation of Fe, Cu, Mn and Zn in the shoots [24]. Similarly, pepper plants exposed to Cd accumulated less Fe, Cu, Mn, Mg and Zn [25]. In turn, in *Suaeda fruticose*, exposure to Cd resulted in increased levels of Mg in the leaves with a simultaneous decrease in the roots. Cd-treated plants also contained lower levels of Na and K. The same study showed that treatment with Cu led to an increased accumulation of K in the shoots [26]. In *Salvia sclarea*, Cd-dependent changes in nutrients varied in different organs. Treatment with this metal led to an increase in Ca, Cu, Fe, Mn and Zn in the roots. The level of Ca, Cu, Mn and Zn was lower in the leaves of a Cd-treated plant when compared to the control. The uptake of Mg was decreased in both the roots and the leaves [27]. Organ- and element-dependent variation in nutrient uptake has been observed even in response to nanomolar Cd concentrations [28]. The leaves of wheat plants exposed to Cr contained lower levels of Ca, Cl, Cu, Fe, K, Mn and Zn [29]. In tomato plants, Pb stress resulted in the attenuated uptake of Cu, Fe, K, N, Mn and Zn [30].

Photosynthesis is one of the most metal-sensitive processes. The impact of metals on photosynthesis has been reviewed in detail by Sharma et al. (2020) [31]. Exposure to metals and metalloids leads to decreased chlorophyll content [32,33,34]. This process depends on changes in the activity of enzymes involved in chlorophyll metabolism. It has been shown that Cd hampers the activity of three enzymes engaged in chlorophyll biosynthesis—δ-aminolevulinic acid, dehydratase and protochlorophyllide reductase [32,33,34]. On the other hand, treatment with Pb at low concentrations led to increased activity of the chlorophyll degrading enzyme chlorophyllase [35]. Several metals, including Cd, Cu, Hg, Ni, Pb and Zn, might substitute Mg in chlorophyll, leading to its alerted functioning [14]. Photosynthesis efficiency might also be decreased due to the metal-dependent inhibition of the electron transfer chain, inhibition of enzymes involved in the light-independent photosynthesis phase and changes in chloroplast ultrastructure [31]. Plants’ exposure to these toxic elements results in the reduced number and size of chloroplasts, reduced grana stacking, changes in starch accumulation and the occurrence of plastoglobuli [36,37,38].

Higher concentrations of metals and metalloids exhibit a genotoxic effect. Studies applying comet assay, which is used to estimate the extent of DNA strand breaks, demonstrated an increased ratio of DNA damage in plants exposed to Al, As, Cd, Co, Cu, Pb, Tl and Zn [39,40]. Metal dependent genomic instability in various plant species has also been confirmed by randomly amplified polymorphic DNA technique (RAPD) [41,42,43]. Another symptom of genotoxicity is the presence of chromosome aberrations. It has been evidenced that metals and metalloids, such as As, Cd, Cr, Cu, Ni, Pb, Zn, cause micronuclei formation, chromosome fragments, bridges, stickiness, c-metaphase and polyploidy [44,45,46,47]. Genotoxicity frequently leads to cell-cycle arrest. Accordingly, the inhibition of cell division reflected by a decrease in mitotic index has been noted in metal-treated plants [44,45]. Hampered cell division might result also from metal-dependent alterations in the cytoskeleton. For instance, it has been evidenced that exposure to Cd results in rearrangements of microtubule formation, attenuated expression of α-, β- and γ-tubulins on protein and/or transcripts level and alterated tubulins’ post-transcriptional modifications in the roots of soybean seedlings [48,49].

All of the described processes result in hampered growth and reduced fitness of metal-stressed plants. Although the direct effect of metals and metalloids is relatively well-studied, there is limited information on plants’ post-metal stress performance. The next section of the article describes reports on the process of plant recovery after metal exposure.

## 3. Post-Stress Recovery

Several studies show that metals can inhibit the process of germination [50]. Research on three plant species belonging to the *Mimosoideae* family, native to North Africa—*Acacia rabbina* and *Acacia tortilis* and invasive *Prosopis juliflora*—confirmed that treatment with Cu and Pb hampered seed germination. However, metal-treated seeds of *P. juliflora* restored their germination capacity after transfer to optimal conditions. A significant increase in the germination rate has been noted after four days of the recovery phase. On the contrary, the seed of *A.tortulis* and *A. rabbiana* showed hampered germination throughout the whole 20 days of the recovery period. These results indicate that seeds of different plant species show distinct capacities for germination after metal treatment [51]. The soybean seeds imbibed for 2 h in Cd solutions and thereafter transferred to optimal conditions showed the same germination rate as the control seeds. The emerged seedlings also showed the same growth and cell viability as the control. However, past treatment with Cd at its highest concentration (25 mg/L corresponding to 223 µM) resulted in a significant decrease of seedling antioxidant activity [52].

The efficiency of plant growth recovery after metal stress depends on the plant species, applied metal, stress intensity and duration. In a study on *Lemna minor*, plants were exposed for three days to Cu, Cd, Ni and Zn at their EC_50_ (effective concentration) respective concentrations and thereafter were transferred for an additional seven days to optimal medium. The results revealed that the most rapid and efficient recovery has been observed in the case of exposure to Ni. The *Lemna minor* plants were capable of full growth restoration already after three days of recovery phase. Similarly, plants previously exposed to Cd and Cu nearly fully restored their growth. The lowest recovery efficiency has been observed in the case of Zn. Even after seven days of cultivation in optimal conditions, the plants reached slightly over 60% of the growth observed in the control. It is worth mentioning that *Lemna minor* plants are characterized by a high growth rate, which might affect its adaptation to metal stress. The time course of the recovery phase was accompanied by a decrease in the internal concentrations of metals, which is explained by the fast growth and consequent internal metal dilution [53]. The study on tobacco suspension cells evidenced that there is a certain threshold beyond which the recovery is no longer possible. Cells treated for three days with Cd at the concentration of 50 µM fully restored their growth after transfer to optimal medium. However, prolongation of the stress conditions by only one day completely enabled the recovery process. No viable cells were detected after four days of recovery phase [54]. The recovery rate of the medical plant *Tetradenia riparia* reached 80% in the case of whole plants and 100% in the case of cuttings derived from Cd-treated plants. However, even after three weeks of cultivation in optimal conditions, root length and leaf elongation of recovered plants was decreased by over twofold when compared to the control [55]. Soybean seedlings were able to restore their growth even after severe short-term Cd stress. All seedlings treated for two days with Cd at the concentration 10 and 25 mg/L, corresponding to 89 and 223 µM, respectively, restored their growth during the seven days of recovery. The only significant difference in growth parameters comprised the shortening of the roots noted in plants previously exposed to a higher metal concentration [56].

Growth restoration is accompanied by the recovery of physiological and biochemical parameters, at least to some extent. Green alga *Scenedesmus* spp. treated with Cu or Zn at two concentrations exhibited alerted nitrogen metabolism reflected by a decrease in NO_3_^−^ uptake and attenuated nitrate reductase (NR) activity. Within 96 h of recovery, phase plants managed to completely restore NR activity. The recovery process was faster in the case of alga previously exposed to lower metal concentrations. Restoration of NR activity was most likely dependent on *de novo* protein synthesis as the application of translation inhibitor (cycloheximide) results in significantly lower enzyme activity throughout the whole recovery phase. On the other hand, NR recovery was accelerated by previous exposure to continuous light. Nitrate uptake reached control levels only in plants subjected to milder stress conditions. In the case of treatment with higher metal concentrations, the NO_3_^−^ uptake came to approximately 65% of the control [57]. In *Tetradenia riparia*, a period of three weeks of recovery resulted in full restoration of net CO_2_ assimilation and partial restoration of stomatal conductance [55]. Previously Cd-treated soybean seedlings showed the same levels of chlorophyll and photosynthesis parameters as the control [56].

Although the above-described examples in general indicate that plants are able to efficiently recover from metal stress and restore their growth and physiological processes, there are also reports showing the long-lasting negative effects of past metal exposure. In rice single, short-term (24 h) treatment with 50 µM Al caused DNA damage, noted even after 15 days of recovery period. Previously Al-treated plants showed a decrease in mitotic and meiotic index and elevated levels of chromosomal abnormalities including chromosome fragments, chromosome clumping and micronuclei formation. These abnormalities were accompanied by increased pollen sterility [58]. Metal-dependent genotoxic effects have been also observed in tobacco suspension cells. Treatment with 50 µM Cd resulted in fragmentation of DNA. However, in this case, withdrawal of metal resulted in the rapid repair of the DNA damage accompanied by an increase in telomerase activity [54].

The described examples show that recovery capacity varies among plant species. For instance, the potential of tobacco suspension cells to recover after Cd is lower than in the case of *Tetradenia riparia* and soybean [53,55,56]. However, further studies on various plant models with differing metals and concentrations would be needed to gain a better view on the recovery potential of specific plant species. It would be particularly interesting to monitor the regeneration process in metal hyperaccumulators. It can be predicted that hyperaccumulators are equipped with mechanisms enabling not only higher metal tolerance but also efficient post-stress recovery.

The reviewed studies, summarized in Table 1, were conducted solely in laboratory conditions. However, research has been carried out also on moss growing naturally in mining areas. In the case of moss, *Scopelophila cataractae* samples were collected from mining sites varying in metal concentration, grown for three months in optimal conditions and thereafter treated with Cd and Cu. Plants obtained from heavily contaminated sites were characterized by higher metal-tolerance, reflected by better growth on metal-supplemented medium [59]. It can be suspected that efficient recovery from metal stress is a universal process observed in various taxons. Indeed, similarly to moss, lichens collected from polluted sites exhibited higher metal-tolerance than their counterparts from unpolluted areas. Peltigera sp samples transferred from mining areas to optimal growing conditions and thereafter subjected to metal stress showed less pronounced Zn-dependent inhibition of nitrogen fixation and photosynthesis rate and attenuated Cd-dependent membrane damage [60]. A more recent study showed that lichens belonging to Hypocenomyce scalaris, Hypogymnia physodes and Lepraria incana species collected from contaminated areas exhibited in general a less intensive decline in chlorophyll level in response to Zn and Pb when compared to lichens from control sites [61].

Several mechanisms likely stand behind efficient recovery. Firstly, the metals/metalloids need to be separated from damage-prone structures through deposition in particular cellular compartments. Cell walls are a major site of metal immobilization. In some plant species, 50 to 70% of take-up metals are deposited in this compartment, whereas cell-wall polysaccharides exhibit the highest metal binding capacity. It is evidenced that in reaction to metals, plants modulate their cell wall composition and structure. These modulations include increasing the levels of polysaccharides such as pectins and callose, accompanied by cell wall thickening. Such modifications might increase the metal binding capacity and facilitate the immobilization of these elements [reviewed in 17]. Metals and metalloids are also sequestrated into vacuoles, either through direct or vesicle transport. Tonoplast are equipped with numerous metal transporters including MTP, CAX, VIT, VTL and ACR transporters and HMA pumps. An important sequestration mechanism includes metal binding by phytochelatins and sequestration into vacuoles through ABC transporters [62].

Secondly, the already caused damage needs to be repaired. The earlier cited study on tobacco suspension cells highlighted the importance of increased telomerase activity for post-metal stress repair of DNA [54]. Other studies showed that in *Medicago truncatula* copper up-regulates the expression of genes encoding proteins associated with RNA and DNA metabolism, namely tyrosyl-DNA phosphodiesterase 2 (Tdp2), transcript elongation factor II-S (TFIIS) and transcript elongation II-S factor-like protein (MtTFIIS-like) [63,64]. The encoded proteins might be engaged in DNA stabilization as they exhibit such functions in animal models. Additionally, the hypothesis is strengthened by the fact that plants over-expressing *Tdp2* showed lower numbers of double strand breaks (DSBs) in their DNA [63]. The symptoms of metal-dependent damage also include alterations in protein folding and structure [12]. The processes of protein folding, re-folding, assembly, stabilization and degradation is mediated by heat shock proteins (HSP) [65]. Induced expression of HSPs in response to metals and metalloids has been observed in various plant species including barley (*Hordeum vulgare*), poplar (*Populus yunnanensis*), duckweed (*Lemna minor*) and alternate watermilfoil (*Myriophyllum alterniflorum*) [66,67,68,69]. It is likely that the observed metal-dependent stimulation of HSP biosynthesis minimizes the damage through protein protection from misfolding and/or involvement in degradation of already misfolded proteins.

Some reports show that the defense mechanisms activated in response to past metal exposure have beneficial effects on plant performance under other stress conditions. This phenomenon, called cross-tolerance, cross-resistance or cross-protection, has been observed, for instance, in the case of Cu-treated pepper plants subsequently infected with *Verticillium dahlia*. Exposure to Cu resulted in increased peroxidase activity, the accumulation of phenolics and induced the expression of four defense-related genes encoding peroxidase, β-1,3-glucanase, sesquiterpene cyclase and pathogen-related gene 1 (PR1). At the same time, Cu-treated plants showed less severe symptoms of *Verticillium dahlia* infection reflected by lower numbers of wilted leaves and a less pronounced reduction of stem length [70]. Similarly, treatment of potato plants with Al led to changes in the level of signaling molecules, including nitric oxide (NO), hydrogen peroxide (H_2_O_2_) and salicylic acid (SA). The modulation of the signaling network was accompanied by the induction of the expression of several pathogen-related genes (PR) and induced activity of β-1,3-glucanase and chitinase. Plants previously exposed to Al were less prone to *Phytophtora infestans* infection, which was evidenced by the lower number and area of leaves showing infection symptoms [71]. The earlier cited studies showing that moss and lichens collected from contaminated sites showed less pronounced symptoms of Cd, Cu, Zn and/or Pb toxicity in laboratory conditions indicates that past exposure to metals might also enhance plants tolerance towards recurrent metal stress [59,60,61].

The described findings evidence that past metal stress leads to relatively lasting changes in the accumulation of specific transcripts and proteins. These changes can influence the future performance of plants. Besides enhanced resistance to pathogens, it has been also shown that exposure to metals alerts the pattern of mineral accumulation. Soybean plants recovered from Cd stress contained higher levels of K, Mg and Mn. The possible explanation for this phenomenon is Cd-driven modulation of the expression and activity of transporters [56]. Imprints induced by stress conditions include changes in histone modifications and DNA methylation level. *Tetradenia riparia* plants showed a decrease in methylation of histone H3 in response to Cd treatment. The authors suggest that the observed lower level of methylation is associated with quiescence strategy in reaction to the stress. This strategy consists in hampered growth and simultaneous maintenance of key cellular processes. Its introduction by plants enables faster and/or more efficient recovery after termination of the unfavorable conditions. Indeed, besides the decrease in the histone H3 methylation, the *Tetradenia riparia* plants were also characterized by slowed down growth, formation of root primordia and the efficient restoration of growth and photosynthesis after transfer to optimal conditions [55]. Alerted DNA methylation levels in response to metals were observed in various plant species, such as Arabidopsis, rice, wheat, lettuce or garden cress. However, no clear pattern of changes could be distinguished. The methylation pattern differed among plant species and applied metal concentration [72,73,74,75,76,77]. Modulation of DNA methylation and histone modifications constitute important component of plant memory, defined as the ability to respond faster and/or more efficiently to reoccurring stress [78,79]. Thus, it would be interesting to explore the possible plant memory of past metal stress. To the best to our knowledge, there are no data available on the topic.

The association between metal toxicity, plants’ defense mechanisms and future plants’ performance including efficient recovery are presented in Figure 1. 

## 4. Conclusions

Plants’ capacity to recover from metal stress depends on the type of applied element, its concentration and the duration of the exposure. Several studies evidence that plants are capable of efficient recovery even from relatively severe stress conditions. The recovery is dependent on various mechanisms including metal immobilization and sequestration and activation of repair systems. The recovery might also be facilitated by the introduction of the quiescence strategy. Past stress might leave in plants relatively long-lasting imprints such as changes in ultrastructure (e.g., modified composition of cell walls), alerted levels of certain transcripts and proteins, modulated mineral uptake capacity, alterations in DNA structure or changes in the level of DNA methylation and histone modifications. The described effects can have adverse or beneficial impacts on future plant fitness. For instance, stable DNA damage is associated with decreased pollen fertility [58]. On the other hand, the accumulation of defense-related proteins generates a higher resistance to pathogens [70,71]. In addition, the studies on moss and lichens imply that chronic exposure to metals leads to adaptation to the contaminated environment and enhancement of metal tolerance [59,60,61].

Getting better insights into processes governing the post-metal stress period might lead to the development of new applications. For instance, it may help in the enhancement of plants’ resistance to pathogens and tolerance towards other stress factors. Properly designed (in terms of the duration and concentrations) pre-treatment with essential metals would lead to cross-resistance and limit pathogen infections in agronomically and industrially important plant species. Pre-exposure to metals could also enhance the metal tolerance in plants used for the decontamination of terrains by the means of phytoremediation. In addition, metal pre-treatment might stimulate the biosynthesis of plant secondary metabolites with pro-healthy activity. Several cases of metal dependent induction of compounds exhibiting strong antioxidant activity have been evidenced [80]. For example, Cu elicited the biosynthesis of new glucoside ester with strong antioxidant properties in *Portulaca oleacera* leaves [81]. In turn, Mn enhanced the biosynthesis of two bioactive compounds—thymol and thymoquinone—in *Nigella sativa* suspension culture [82].

Further research on the recovery after metal stress could also contribute to the redefinition of metal tolerance. As described in the introduction, so far, tolerance is perceived mainly as efficient growth under metal exposure. However, some plant species introduce a quiescence strategy to save carbohydrate reserves for rapid growth after stress decline. Thus, the introduction of a recovery phase into toxicity tests might provide a better view on the exact metal tolerance and regeneration potential of plants.

## Figures and Tables

**Figure 1 plants-10-00450-f001:**
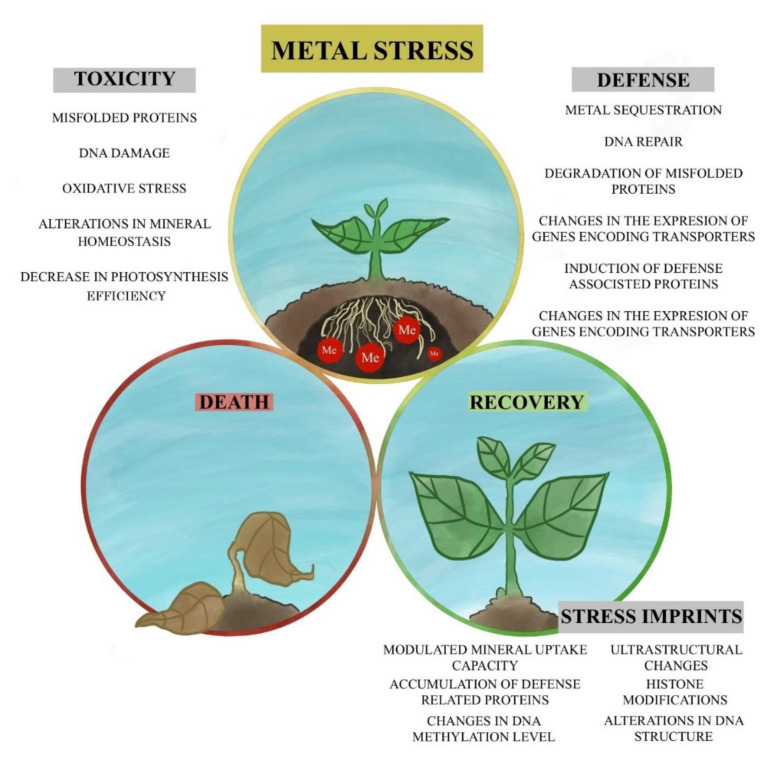
Metal toxicity might cause severe injuries and result in plants death. On the other hand, exposure to metals stimulates defense mechanisms, which enable plant survival and efficient recovery after stress termination. Past stress conditions might leave relatively stable imprints that influence future plant performance in negative or positive manner. The figure has been prepared by Jorge Granados-Tello (jorgegranadostello@gmail.com) and is published with his permission.

**Table 1 plants-10-00450-t001:** The list of studies on plants recovery from metal stress with indicated plant species, applied metal and duration of the treatment and recovery period.

Metal/Concentration	Treatment Duration	Plant Species	Recovery Time	References
Cd (50 µM)	up to 4 days (96 h)	Tobacco suspension cells (*Nicotiana tabacum*)	4 days	54
Cd (30 and 150 µM)	35 days (5 weeks)	*Tetradenia riparia*	21 days	55
Cd (10 and 25 mg/L; 89 and 223 µM)	2 days (48 h)	Soybean (*Glycine max*)	7 days	56
Zn (1026.4 µM) Cu (12.1 µM) Ni (81.8 µM) Cd (3.5 µM)	3 days (72 h)	Duckweed (*Lemna minor*)	7 days	53
Cu (1000 and 2000 ppm) Pb (1000 and 2000 ppm) Cu + Pb (500 and 1000 ppm each metal)	20 days	Three species belonging to the *Memosoideae* family: *Acacia tortilis*, *Acaica raddiana*, *Prosopis juliflora*	20 days	51
Cu (0.5–50 μM), Zn (1–100 μM)	2 days (48 h)	Green alga *Scenedesmus* sp.	4 days	57
Al (50 µM)	1 day (24 h)	Rice (*Oryza sativa*)	15 days	58

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
