# Peer review of "Plant Recovery after Metal Stress—A Review"

_plants, 2021, doi:10.3390/plants10030450_

Round 1

Reviewer 1 Report

The manuscript, a review, summarizes results, concerning the recovery of plants after expositions to heavy metals. The subject appears to be interesting not only because the recovery after the metabolic change induced by heavy metal treatments is interesting for plant physiology study but also because these treatments appropriately set up can be used to produce an adaptation of plant to heavy metals. This possibility, even if it is presented and discussed in the paper, is not conveniently introduced and discussed between the aim of the paper.

In the present version the manuscript needs changes and improvement. In particular

-line 11: “post metal stress”. This should be clarified and suggested that the reduction of the presence of heavy metal is obtained experimentally in most cases but also exceptionally naturally.

(see line 53 and following)

-line 57-59. These sentences are not clear: what means “energy” must be specified.

-Fig 1 spelling : efficiency and not eficiency

Author Response

We would like to thank the reviewers for their comments and suggestion, which resulted in significant improvement of the manuscript. All modifications included in the manuscript are marked in blue colour. The main changes included:

  • Change of the title
  • Extension of the abstract by clarification of the term “post metal stress” and relating the process to natural conditions
  • Expansion of the aim of the manuscript by metal induced adaptation processes
  • Inclusion of the information on enhanced metal tolerance of moss and lichens collected from contaminated sites when compared with samples collected from unpolluted regions
  • Proposing of the extension of the definition of plant “tolerance”
  • Extending and modification of the “conclusions” section by putting emphasis on the scientific and agronomical significance of studies on the post metal stress period
  • Expansion of the references included in the text, in particular in the “Metal toxicity in plants” section
  • Adjustment of citation and reference list format

Comments and Suggestions for Authors

The manuscript, a review, summarizes results, concerning the recovery of plants after expositions to heavy metals. The subject appears to be interesting not only because the recovery after the metabolic change induced by heavy metal treatments is interesting for plant physiology study but also because these treatments appropriately set up can be used to produce an adaptation of plant to heavy metals. This possibility, even if it is presented and discussed in the paper, is not conveniently introduced and discussed between the aim of the paper. 

We would like to thank for highlighting of the important issue of the impact of past metals stress on plants adaptation. We added information on this phenomenon in the abstract and introduction (lines 19-22; 58-59).

In the present version the manuscript needs changes and improvement. In particular

-line 11: “post metal stress”. This should be clarified and suggested that the reduction of the presence of heavy metal is obtained experimentally in most cases but also exceptionally naturally.

(see line 53 and following)

The term “post metal stress conditions” has been clarified in the abstract and related to  natural conditions (lines 15-19).

-line 57-59. These sentences are not clear: what means “energy” must be specified.

The term has been specified as carbohydrates reserve.

-Fig 1 spelling : efficiency and not eficiency

The word has been corrected.

Reviewer 2 Report

  • The title of the manuscript is more appropriate for a popular magazine than for a scientific one
  • The described phenomenon is not traditional and is supposed to be typical for all Earth flora. Taking this into account, one may suppose the existence of similar responses for different species including glycophytes and halophytes, hyperaccumulators, secondary accumulators (or indicators) and non accumulators, aquatic and terrestrial plants and even mushrooms, lichens and mosses. It is highly desirable to include the available data on these objects into the review.
  • Taking into account Lemna minor data (plant with extremely high growth rate) a question arises about the effect of metabolism intensity on plants response to metals loading and tolerance to metals supply- add discussion
  • In the discussion of the term ‘tolerance’ the authors indicate that the explanation (changes in growth parameters straight after exposure) is not accurate but they do not propose any other definition
  • It is desirable to indicate more widely the prospects of this phenomenon utilization

Author Response

We would like to thank the reviewers for their comments and suggestion, which resulted in significant improvement of the manuscript. All modifications included in the manuscript are marked in blue colour. The main changes included:

  • Change of the title
  • Extension of the abstract by clarification of the term “post metal stress” and relating the process to natural conditions
  • Expansion of the aim of the manuscript by metal induced adaptation processes
  • Inclusion of the information on enhanced metal tolerance of moss and lichens collected from contaminated sites when compared with samples collected from unpolluted regions
  • Proposing of the extension of the definition of plant “tolerance”
  • Extending and modification of the “conclusions” section by putting emphasis on the scientific and agronomical significance of studies on the post metal stress period
  • Expansion of the references included in the text, in particular in the “Metal toxicity in plants” section
  • Adjustment of citation and reference list format

Comments and Suggestions for Authors

  • The title of the manuscript is more appropriate for a popular magazine than for a scientific one

The title has been changed to a more appropriate for a scientific journal.

  • The described phenomenon is not traditional and is supposed to be typical for all Earth flora. Taking this into account, one may suppose the existence of similar responses for different species including glycophytes and halophytes, hyperaccumulators, secondary accumulators (or indicators) and non accumulators, aquatic and terrestrial plants and even mushrooms, lichens and mosses. It is highly desirable to include the available data on these objects into the review.

We also suspect that recovery is an universal process observed in various groups of plant and other organisms. However, the literature data on the topic is still very limited. We failed to find related study concerning glycophytes, halophytes, hyperaccumulators. However, we have included information on the prospects of such studies in the “Post-stress recovery” section (lines 212-216).

We have also included information from three studies carried out on moss and lichens (lines 217-230). We thank for the comment as we feel that inclusion of this three studies added important information and greatly improved the manuscript.

  • Taking into account Lemna minor data (plant with extremely high growth rate) a question arises about the effect of metabolism intensity on plants response to metals loading and tolerance to metals supply- add discussion

The information of the possible impact of Lemna minor fast growth on the metal response had been included (lines 167-171).

  • In the discussion of the term ‘tolerance’ the authors indicate that the explanation (changes in growth parameters straight after exposure) is not accurate but they do not propose any other definition

The suggestion on extension of the tolerance tests by inclusion of the recovery phase has been included in the text in introduction (lines 67-69) and conclusions (lines 329-334).

  • It is desirable to indicate more widely the prospects of this phenomenon utilization

The possible applications of studies on post metal stress phase have been introduced in the “Conclusion” section (lines 317-328).

Reviewer 3 Report

Comments to authors

The review mansucript presents an overview of first responses of plants to heavy metals stresses. In the second and the principal part, authors describe the recoveny of plant growth after metals stresses.

The manuscript is well written and well documented. Moreover, as presented by authors the plant growth recovery is largely less documented that plant reaction to heavy metals stress

This manuscript meets the expectantions of Plants.

Nevertheless, some modifications are necessary before considering it for publication.

As for other stresses, some plant species are more able to revover the growth. This part is not presented (except studies cited in table 1).

Is there any interest, agronomically and practically senses, to know the growth recovery after metal stress ? This is lacking.

What about Brassicaceae, more known to accumulate heavy metals ? (this question is pure scientific curiosity) If no study, you answer is sufficient.

Conclusion

This part needs to be expanded.

P9 L304-307. The sentences should be rewritten. It seems more judicious to present prespectives (scientific and agronomic…), but not the future paper of the authors.

Minor remark

Please follow the authors guidelines of the Journal for references indexation bot the the text and in the liste of references.

Author Response

We would like to thank the reviewers for their comments and suggestion, which resulted in significant improvement of the manuscript. All modifications included in the manuscript are marked in blue colour. The main changes included:

  • Change of the title
  • Extension of the abstract by clarification of the term “post metal stress” and relating the process to natural conditions
  • Expansion of the aim of the manuscript by metal induced adaptation processes
  • Inclusion of the information on enhanced metal tolerance of moss and lichens collected from contaminated sites when compared with samples collected from unpolluted regions
  • Proposing of the extension of the definition of plant “tolerance”
  • Extending and modification of the “conclusions” section by putting emphasis on the scientific and agronomical significance of studies on the post metal stress period
  • Expansion of the references included in the text, in particular in the “Metal toxicity in plants” section
  • Adjustment of citation and reference list format

Comments to authors

The review mansucript presents an overview of first responses of plants to heavy metals stresses. In the second and the principal part, authors describe the recoveny of plant growth after metals stresses.

The manuscript is well written and well documented. Moreover, as presented by authors the plant growth recovery is largely less documented that plant reaction to heavy metals stress

This manuscript meets the expectantions of Plants.

Nevertheless, some modifications are necessary before considering it for publication.

As for other stresses, some plant species are more able to revover the growth. This part is not presented (except studies cited in table 1).

The information on possible variation in plant recovery potential has been added in the manuscript (lines 210-216).

Is there any interest, agronomically and practically senses, to know the growth recovery after metal stress ? This is lacking.

The appropriate information has been added to the “conclusions” section (lines 317-328)

What about Brassicaceae, more known to accumulate heavy metals ? (this question is pure scientific curiosity) If no study, you answer is sufficient.

Unfortunately we did not find studies on Brassicaceae. We have added information on the prospects and benefits of studies on metal accumulators in “Post-stress recovery” section (lines 212-216)

Conclusion

This part needs to be expanded.

P9 L304-307. The sentences should be rewritten. It seems more judicious to present prespectives (scientific and agronomic…), but not the future paper of the authors.

We have extended and modified the “Conclusions” section. Thank you for the remark as in our opinion the section has been significantly improved.

Minor remark

Please follow the authors guidelines of the Journal for references indexation bot the the text and in the liste of references.

We have changed the citation and reference list according to the guidelines.

Reviewer 4 Report

The main purpose of this manuscript is to review information on plant recovery after metal stress and possible mechanisms behind effective post metal stress regeneration. The section “ Metal toxicity in plants “ briefly described the mechanisms of metal toxicity. However, this part is written descriptively with very little and even without any citations. The reader must trust the statements of the authors.

  • For example, in many sentences and statements (lines 87-90 and 105-115) the authors do not cite or point any references. The same applies to paragraphs on lines: 140-145, 155-161, 180-189, 210-216, etc.
  • In the section about the uptake of some elements and nutrients after Cd treatment (lines 93-101), only 2 old references are cited, although there are many recent articles in last years on this.
  • On lines 161 and 163: instead of only "plants" indicate plant species.
  • References must be numbered in order of appearance in the text in square brackets [ ]. (see Guides for authors).
  • The reference list must be prepared in the same style as required by the Journal (see also Guides for authors).

Author Response

We would like to thank the reviewers for their comments and suggestion, which resulted in significant improvement of the manuscript. All modifications included in the manuscript are marked in blue colour. The main changes included:

  • Change of the title
  • Extension of the abstract by clarification of the term “post metal stress” and relating the process to natural conditions
  • Expansion of the aim of the manuscript by metal induced adaptation processes
  • Inclusion of the information on enhanced metal tolerance of moss and lichens collected from contaminated sites when compared with samples collected from unpolluted regions
  • Proposing of the extension of the definition of plant “tolerance”
  • Extending and modification of the “conclusions” section by putting emphasis on the scientific and agronomical significance of studies on the post metal stress period
  • Expansion of the references included in the text, in particular in the “Metal toxicity in plants” section
  • Adjustment of citation and reference list format

Comments and Suggestions for Authors

The main purpose of this manuscript is to review information on plant recovery after metal stress and possible mechanisms behind effective post metal stress regeneration. The section “ Metal toxicity in plants “ briefly described the mechanisms of metal toxicity. However, this part is written descriptively with very little and even without any citations. The reader must trust the statements of the authors.

  • For example, in many sentences and statements (lines 87-90 and 105-115) the authors do not cite or point any references. The same applies to paragraphs on lines: 140-145, 155-161, 180-189, 210-216, etc.

We have included the references and changed the form of references in the text and in the reference list.

lines 87-90: The references have been expanded and included in the text. 

lines 105-115: photosynthesis

In the case of lines 140-145; 155-161; 180-189 and 210-216 the fragments are based on individual references described in more detail. The mentioned references are cited after each fragment (lines 140-145: Chmielowska-Bąk et al. 2020, lines 155-161: Bazihizina et al 2016, lines 180-189: Mohanty et al. 2004; Fojtová et al. 2002, lines 210-216: Faé et al. 2014).

  • In the section about the uptake of some elements and nutrients after Cd treatment (lines 93-101), only 2 old references are cited, although there are many recent articles in last years on this.

We have included additional references in the fragment concerning the impact of metals on nutrient uptake (references 24-30). The fragment concerning the impact of Cd on plants mineral composition has been significantly expanded (lines 102-111).

  • On lines 161 and 163: instead of only "plants" indicate plant species.

The plant species has been introduced.

  • References must be numbered in order of appearance in the text in square brackets [ ]. (see Guides for authors).

The citations where corrected everywhere in the text.

  • The reference list must be prepared in the same style as required by the Journal (see also Guides for authors).

The reference list has been corrected.

Round 2

Reviewer 4 Report

After the corrections, the manuscript can be accepted for publication.